# MeRCI: A new metric to evaluate the correlation between predictive uncertainty and true error

## Abstract

As deep learning applications are becoming more and more pervasive, the question of evaluating the reliability of a prediction becomes a central question in the machine learning community. This domain, known as predictive uncertainty, has come under the scrutiny of research groups developing Bayesian approaches to deep learning such as Monte Carlo Dropout. Unfortunately, for the time being, the real goal of predictive uncertainty has been swept under the rug. Indeed, Bayesian approaches are solely evaluated in terms of raw performance of the prediction, while the quality of the estimated uncertainty itself is not assessed. One contribution of this article is to draw attention on existing metrics developed in the forecast community, designed to evaluate both the sharpness and the calibration of predictive uncertainty. Sharpness refers to the concentration of the predictive distributions and calibration to the consistency between the predicted uncertainty level and the actual errors. We further analyze the behavior of these metrics on regression problems when deep convolutional networks are involved and for several current predictive uncertainty approaches. A second contribution of this article is to propose an alternative metric that is more adapted to the evaluation of relative uncertainty assessment and directly applicable to regression with deep learning. This metric is evaluated and compared with existing ones on a toy dataset as well as on the problem of monocular depth estimation.

## 1 Introduction

Nowadays, we are able to teach machines a wide and impressive variety of tasks. With the outcome of deep learning these past few years, we are also capable of reaching - and even sometimes surpass - human performances, *e.g.* for vision tasks (Kokkinos, 2015; Lu & Tang, 2015). However, although deep learning allows us to attain great performance, it remains imperfect and we still lack the means to assign a confidence to the predictions of a trained network. While the robustness of deep learning algorithms to adversarial attacks has been very actively addressed in the recent literature (see *e.g.* (Papernot et al., 2016)), the question of predicting the uncertainty of predictions made by deep neural networks has received much less attention. Yet, for countless applications it is a matter of critical importance to know whether the estimate output of the algorithm is reliable or not.

In this paper we take the example of depth estimation from single monocular images (see *e.g.* (Liu et al., 2016)). For such a task, estimation algorithms are trained to regress a depth map from an input RGB image. If we want to use such algorithms as a measurement tool, it is required to have, in addition to the predicted depth, an uncertainty map providing an estimate of the accuracy for each pixel. Uncertainty prediction is indeed a very valuable information, but it depends on the content of the images. It is likely that depth estimation on textureless regions will be less accurate, or, that dark images lead to less accurate estimations than well-exposed images.

In the literature of deep learning, some recent techniques have emerged to estimate uncertainties along with the network's predictions (Gal & Ghahramani, 2016; Lakshminarayanan et al., 2017; Kendall & Gal, 2017). However, these predictive uncertainties are evaluated either by their influence on the predictions performance, or qualitatively by visually checking the uncertainty maps. This situation is in contrast with other domains such as forecasting that have established standard tools

to evaluate predictive uncertainty (Uusitalo et al., 2015; Gneiting et al., 2007; Gneiting & Raftery, 2007). We believe that these evaluation metrics should benefit the computer vision literature to evaluate the quality of the uncertainty estimate itself.

One of the contributions of this paper is a comprehensive review of the existing metrics that allow to evaluate the quality of predictive uncertainty. Another contribution is an in-depth analysis of the behavior of these metrics on regression problems, when deep convolutional networks are involved and for several current predictive uncertainty approaches. Finally, this paper proposes a novel evaluation metric that is more adapted to the evaluation of relative uncertainty assessment and directly applicable to the current practice in deep learning. This metric is evaluated and compared to the reviewed ones on a toy dataset as well as in the context of monocular depth estimation from images.

## 2 RELATED WORK

Making uncertainty estimates along with tasks predictions is becoming more and more crucial to the field of deep learning. To this end, a common path is to transform deterministic neural networks, in which each parameter is a scalar, into probabilistic Bayesian models, where parameters are replaced by probabilty distributions. Because the inference becomes intractable, several papers rely on approximate solutions. Among the most popular techniques, we count Probabilistic BackPropagation (PBP) (Hernández-Lobato & Adams, 2015), Monte Carlo Dropout (Gal & Ghahramani, 2016), but also some non-Bayesian ones such as Deep ensembles (Lakshminarayanan et al., 2017) or the learned attenuation loss proposed by Kendall & Gal (2017) to capture the aleatoric uncertainty in the data.

However, this literature still lacks proper metrics to evaluate the quality of the uncertainties. Indeed, systematic assessment of the uncertainty estimate itself is largely underestimated in the deep learning community. It is usually the quality of the prediction which is measured, and not the quality of the estimated uncertainty. Concerning the qualitative results, it is common to see maps of the estimated uncertainty displayed as images and visually evaluated. Yet, the literature on deep neural networks still lacks specific methods for the quantitative evaluation of the uncertainty estimates.

In contrast, evaluating predictive uncertainty has been a concern for decades in other fields, leading to the development of well established metrics. The evaluating predictive uncertainty challenge (PASCAL Challenge), Quinonero-Candela et al. (2006) proposed two metrics adapted to regression tasks, that are still commonly used. The normalized MSE aims at evaluating the quality of a prediction and uses the empirical variance of the test estimates to normalize the scores. This is therefore not a direct way to evaluate the predictive uncertainty. The second one, the average negative log predictive density (NLPD), directly takes into account the estimated uncertainty and penalizes both over and under-confident predictions.

A more in-depth treatment can be found in the forecasting literature. Uusitalo et al. (2015) draw up an overview of methods to estimate uncertainty with deterministic models such as: expert assessment, where an expert in the area rates the confidence of a method using his knowledge; model sensitivity analysis, determining how the output response of the model change w.r.t. the inputs and model's parameters; model emulation, consisting of approximating a complex model by a lower-order function, to then apply more easily a sensitivity analysis; spatiotemporal variability, using the spatiotemporal variability as an estimate of the variance; multiple models, where an ensemble of models are used to describe the same domain; and finally data-based approaches, where enough data are available to directly assess the statistical uncertainty related to the model outputs.

Gneiting et al. (2007) propose to evaluate a predictive uncertainty forecast in terms of calibration and sharpness. Calibration refers to the statistical consistency between the predictive distributions and the observations; sharpness refers to the concentration of the predictive distributions. Several graphical tools are proposed to visually evaluate if the probabilistic forecasts are calibrated and sharp. More recently, Kuleshov et al. (2018) propose a method to recalibrate regression algorithms so that the uncertainty estimates lie in the expected confidence interval. They also propose metrics to assess both calibration and sharpness of the predictive distribution. However, they consider them separately while we were driven by the will of assessing both these properties jointly following the tracks of Gneiting *et al.*, using scoring rules (Gneiting & Raftery, 2007). We are therefore interested in the scoring rules, that assess the quality of a forecast, maximizing sharpness subject to calibration.

However, all these metrics imply that the prediction is a probability distribution, which is seldom true in practice. Instead, existing deep learning methods usually predict uncertainty as being a scalar representing the variance of some predictions (*e.g.* with Monte Carlo sampling). We can easily convert prediction/uncertainty pairs to a probability distribution but we therefore have to choose a prior on the distribution. We assess that within reasonable variations, this prior is not of critical importance since it does not change drastically the ranking between different uncertainty predictions, as we will see later.

For these reasons, we propose a novel metric to evaluate prediction/uncertainty couples. We call our metric the **Mean Rescaled Confidence Interval (MeRCI)**. It is based on the calculation of a scaling coefficient, for the interval centered on the prediction to encompass the ground-truth observation. It needs no prior on the distribution, allows to effectively rank different methods of estimations and is parametrized to be robust to outliers.

## 3 COMMON PERFORMANCE METRICS FOR UNCERTAINTY PREDICTION

We assume a regression settings with a training set of $N$ samples denoted as $\mathcal{T} = \{(x_i, y_i^*)\}$ where $x_i \in \mathbb{R}^d$ is the input vector of dimension $d$ and $y_i^* \in \mathbb{R}$ the scalar value to be predicted. In a standard deterministic regression framework, the regressor is a function defined as $f_\theta : \mathbb{R}^d \to \mathbb{R}$, where $f_\theta(x)$ is an estimator of the true function $f(x)$, and $\hat{y} = f_\theta(x)$ is the prediction made by the model. Learning such a regressor consists in finding the parameters $\theta$ minimizing an empirical loss defined by $\frac{1}{N}\sum_{i=1}^{N} L(f_\theta(x_i), y_i^*)$ where $L(\cdot, \cdot)$ accounts for the proximity of its two arguments (e.g. a squared difference). In a probabilistic framework, we predict a whole distribution $P_y$ instead of a deterministic prediction. This distribution has a probability density function $p_y(x)$ and a cumulative distribution $F_y$. In practice, $P_y$ is often characterized by its mean value, which is usually used as the prediction $\hat{y}$, and by its standard deviation $\sigma$. Different approaches to perform such kind of predictions shall be reviewed in section 5.

In the literature of forecasting, we can find several metrics for assessing the quality of a predictive distribution, or forecast. We present here the five most popular ones. All of them are computed as an empirical expectation :

$$\frac{1}{N}\sum_{i=1}^{N} S(P_{y_i}, y_i^*) \tag{1}$$

where the function $S$ relates to the harmony between the prediction $P_y$ and the actual value $y^*$. Depending on the variant under consideration, it may account for sharpness, calibration or both. In the following paragraphs we will refer to $S$ as the adequacy function.

### 3.1 COVERAGE

The coverage is defined for a fixed confidence level $\alpha$ (*e.g.* 95%) as the proportion of observations that actually belong to the associated confidence interval. Denoting $CI_\alpha(P_{y_i})$ the confidence interval of level $\alpha$ for the forecast, then the adequacy function is just the indicator function of the confidence interval:

$$S_{cov}(P_{y_i}, y_i^*) = \mathbb{1}_{\{y_i^* \in CI_\alpha(P_{y_i})\}} \tag{2}$$

The coverage is minimum (0%) if none of the values are in the corresponding confidence interval, and can reach 100% if all the observations are in the interval. As a result, the coverage only accounts for the calibration, and can be arbitrarily improved by making non-sharp predictions (in other words by being pessimistic on purpose).

### 3.2 STRICTLY PROPER SCORING RULES

In the forecast literature, adequacy functions that are positively oriented are known as scoring rules (Gneiting & Raftery, 2007). A scoring rule is said strictly proper if in average, it singles out the true distribution as the optimal forecast in the following sense:

$$\mathbb{E}_{Y \sim P_y^*}[S(P_y^*, Y)] < \mathbb{E}_{Y \sim P_y^*}[S(P_y, Y)] \tag{3}$$

for any forecast $P_y$ different from the ground-truth distribution $P_y^*$. Three common and popular strictly proper scoring rules are:

$$S_{Quadratic}(P_y, y^*) = 2p_y(y^*) - \int_{-\infty}^{+\infty} p_y^2(x)dx \tag{4}$$

$$S_{Logarithmic}(P_y, y^*) = \log(p_y(y^*)) \tag{5}$$

$$S_{Spherical}(P_y, y^*) = \frac{p_y(y^*)}{\sqrt{\int_{-\infty}^{+\infty} p_y^2(x)dx}} \tag{6}$$

## 3.3 CRPS

Another popular metric is the Continuous Ranked Probability Score (CRPS) (Gneiting & Raftery, 2007). The CRPS adequacy function between the forecast $P_y$ (with cumulative distribution function $F_y$) and $y^*$ is defined as:

$$S_{crps}(P_y, y^*) = \int_{-\infty}^{+\infty} \left( F_y(x) - H(x - y^*) \right)^2 dx \tag{7}$$

where $H(t) = \mathbb{1}_{\{t>0\}}$ is the Heaviside step function .

The CRPS is expressed in the same unit as the observed variable and reduces to the mean absolute error if the forecast is deterministic. It is lower bounded by 0 and unlike the previous metrics, it is oriented negatively.

## 4 PROPOSED METRIC

The previous section introduces several metrics for assessing the quality of a forecast. To be applicable, they require the prediction to be defined through a distribution. However, most deep learning regression methods merely output a scalar value $\hat{y}$ along with a standard deviation $\sigma$. Therefore, assessing the quality of these methods with existing metrics depends on the choice of a distribution centered around the predicted value and with the correct variance. By simplicity, one could use a Gaussian prior, but this choice is completely arbitrary. Here we present a metric that wriggles from this difficulty by establishing a score based solely on the parameters $\hat{y}$ and $\sigma$ estimated with the regressor.

We aim for a metric that reflects the correlation between the true error and the estimated uncertainty. Further, we want this metric to be scale independent and robust to outliers. Inspired by the notion of calibration and with scale invariance in mind, we propose to rescale all the $\sigma_i$'s by a common factor $\lambda$ so that the rescaled uncertainties $\tilde{\sigma}_i = \lambda\sigma_i$ are compatible with the observed errors, that is to say $|\hat{y}_i - y_i^*| \le \tilde{\sigma}_i$. Robustness to outliers is achieved by relaxing this constraint such that it is verified by only a certain number of the samples. In practice, we choose to use $95\%$ of inliners to compute the MeRCI score. We show in section 6 that it is a good trade-off for a robust yet representative MeRCI score. In other words, we look for the smallest factor $\lambda^{95}$ such that $95\%$ of the observed errors are less than their corresponding rescaled predicted uncertainty.

Then, the Mean Rescaled Confidence Interval that we propose is an average uncertainty, performed over the $\lambda$-rescaled estimated uncertainties and averaged over the number of evaluated points $N$:

$$MeRCI = \frac{1}{N}\sum_{i=1}^{N} \lambda^{95}\,\sigma_i \tag{8}$$

To obtain an efficient computation of the scaling factor, we first evaluate all ratios $\lambda_i = \frac{|\hat{y}_i - y_i^*|}{\sigma_i}$ and then extract the $95^{\text{th}}$ percentile which corresponds exactly to the desired value.

The MeRCI is inherently insensitive to scaling and allows us to compare methods that predict very different ranges of uncertainties. Besides, a range of relevant values can easily be exposed for this metric. On one hand, the MeRCI score is always lower bounded by the corresponding score for an ideal uncertainty prediction called the **oracle**. This oracle merely predicts an uncertainty equal

to the actual observed error, which yields to $\lambda^{95} = 1$ and a MeRCI score reducing to the Mean Absolute Error. On the other hand, any uncertainty prediction that performs worse than a **constant** uncertainty predictor (i.e. a non-informative predictor) is clearly not worth pursuing. As a result, the MeRCI score obtained by the constant predictor provides a form of upper bound.

In its construction, the MeRCI score follows the track outlined by Gneiting et al. (2007). It reflects the best possible sharpness under the constraint that the uncertainty estimate is calibrated, in the sense that it is consistent with $95\%$ of the samples. Also, the oracle is the sharpest possible uncertainty predictor once the estimated values $\hat{y}_i$ are fixed.

## 5 PREDICTIVE UNCERTAINTY METHODS

We consider several approaches to perform regression tasks with predictive uncertainty. First of all, we take the most used and popular methods in the literature of deep learning, among which we find Bayesian techniques. Due to its small overhead, **Monte Carlo Dropout (MCD)** (Gal & Ghahramani, 2016) is one such technique. At test time, the dropout is kept activated to perform Monte Carlo sampling. The final prediction is the average of the different estimates and their standard deviation is used as a measure of uncertainty.

We also explore some popular ensemble techniques. In ensemble learning, multiple algorithms are used to obtain a predictive performance. Here, the different deterministic outputs are combined together (usually averaged) to obtain the final prediction and their standard deviation is associated to the degree of uncertainty of the prediction. A classical ensemble approach consists in training several networks on the same task but with different initial random seeds. It is the approach employed in Deep Ensembles (Lakshminarayanan et al., 2017), and we refer to this method as **Multi Inits (MI)**. Another popular ensemble way to estimate uncertainty is **Bagging** (Breiman, 1996). It involves using several subsets of the dataset to train several networks.

Then, inspired by state-of-the-art, we also propose several techniques to predict uncertainty in a deep learning framework. Firstly, we propose to consider the network at different stages of the training as being as many different networks as there are epochs. Assuming that, we can simulate an ensemble method coming with no additional cost, with standard training procedures. However, unlike snapshot ensembles (Huang et al., 2017), we do not use a particular learning rate policy during the learning stage. Moreover, we aim to use this kind of ensemble technique seeking more to produce a meaningful uncertainty map than enhancing the overall prediction. We refer to this method as **Multi Epochs (ME)**. On a par with multiple initializations, we propose to use several architectures and refer to this ensemble technique as **Multi Networks (MN)**. Here again, we average the outputs to obtain the prediction and use their standard deviation as the measure of uncertainty. Finally, we think that because we want to measure the correlation between the uncertainty and the error, we can try to directly learn the error of a trained network with another one. The output of this network that learns the error will be therefore the uncertainty estimate paired with the prediction of the initial network. It is referred as **Learned Error (LE)**.

## 6 EXPERIMENTAL VALIDATION: METRICS BEHAVIOR ON A TOY DATASET

In this section, we want to understand how the different metrics we have presented behave on practical machine learning problems, especially on regression tasks. In order to get an intuition on this question, we start with a simple toy example on which we analyze the metrics behavior.

### 6.1 EXPERIMENTAL PROTOCOL

The toy dataset corresponds to a simple one-dimensional regression task. The data consists of 20 training examples, sampled uniformly in the interval $[-4; 4]$. Their corresponding targets are generated as $y = x^3 + \epsilon$, where $\epsilon \sim \mathcal{N}(0, 3^2)$. We add a high bias to each training point in the interval $[-2.3; -1.3]$ to simulate some outliers, as illustrated in Figure 1. A similar dataset has already been used to qualitatively evaluate predictive uncertainty in Hernández-Lobato & Adams (2015) and Lakshminarayanan et al. (2017).

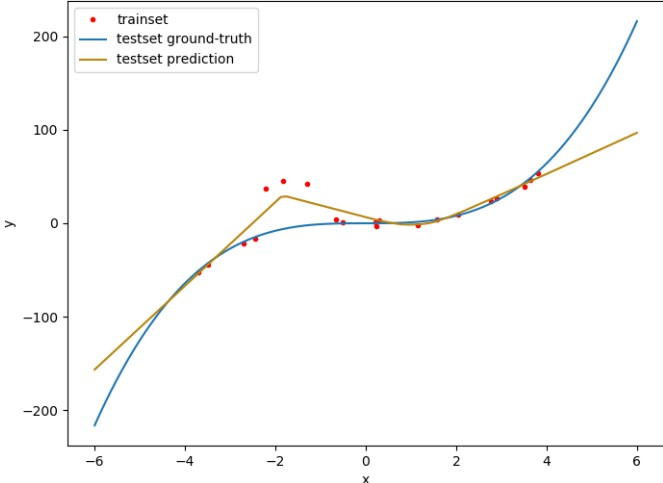

Figure 1: The toy dataset.

Given $x$, we learn to regress $y$ using a neural network with one hidden layer consisting of 100 neurons. We use the ReLU non-linearity and a dropout with dropping probability of 0.2 on the hidden layer. The network predictions on the testset are also illustrated in Figure 1. As expected, we see that the predictions are less accurate outside of the training set as well as in the outliers interval. We are interested in the comparison of several uncertainty estimations techniques. To make it possible, we keep the network predictions fixed in all our experiments. From there, we vary how the uncertainty estimations are calculated with 4 different uncertainty estimation techniques: Multi Inits, Bagging, Monte Carlo Dropout and Multi Epochs.

We will use this toy dataset to discuss how the different metrics are biased to optimistic and pessimistic uncertainty estimations; how robust they are to outliers; and finally verify that they correctly rank the evaluated uncertainty estimation techniques.

## 6.2 SENSIBILITY TO OPTIMISM AND PESSIMISM

We start by studying the simple case of a constant uncertainty estimator, always giving the same uncertainty value $\sigma$ regardless of the data. To do so, we plot the evolution of each metric as $\sigma$ evolves from an overly optimistic setting to an overly pessimistic one. In practice, we choose $\sigma$ in $[0; 100]$ (Figure 2). We see that the classical metrics are greatly impacted by the scaling of $\sigma$, displaying large variations between the extreme cases. We observe that, notwithstanding the considered metric, the optimum value of $\sigma$ is rather low, meaning that it depends more on the method's calibration than on the metric itself.

As expected, we confirm that the coverage is only sensitive to calibration. As such it can be arbitrarily enhanced by increasing $\sigma$. Since it does completely ignore the sharpness, we shall not use it anymore in the rest of the paper. CRPS is rather sensitive to sharpness since its score keeps worsening as $\sigma$ increases. The other scoring rules react more strongly to calibration. In turn, we see that MeRCI score is constant with respect to $\sigma$, and is therefore unbiased to overly optimistic or pessimistic forecasts. Such invariance property is available by design in MeRCI, because we consider that in practice the estimated uncertainties merely cast an insight on the relative magnitude of errors.

## 6.3 ROBUSTNESS TO OUTLIERS

Unlike classical metrics, MeRCI can be tailored towards obtaining more or less tolerance to erroneous data by tuning the percentile value. To gain better understanding, we plot the evolution of MeRCI with respect to the chosen percentile in Figure 3. For comparison purposes, we made 1000 independent runs for each method and we display the median MeRCI evolution curve. In this toy dataset, the ratio of corrupted data is known in advance (and is marked with a vertical red line in the figure). As expected, the evolution of the MeRCI score increases smoothly as the number of

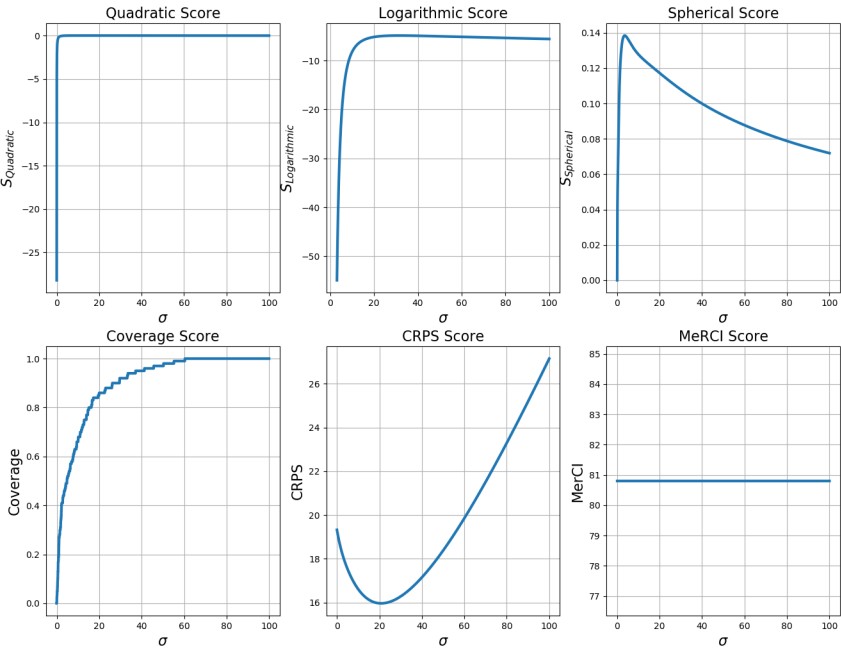

Figure 2: Evolution of scores for each metric according to a given constant uncertainty estimator

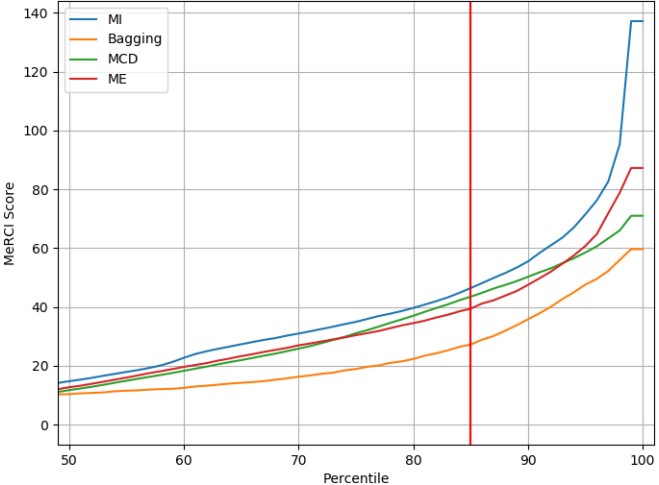

Figure 3: Evolution of the MeRCI score for each method based on the chosen percentile. The red line marks the actual percentage of outliers

expected inliners increases. Besides, MeRCI is robust to outliers since it ranks the different uncertainty estimation techniques in the same order given sufficient data (here from the $74th$ percentile) and until the outliers upset the scores with too much weight in the final decision ($92th$ percentile). In practice, the percentile should be chosen according to the number of supposed outliers. Yet it is also noticeable that knowing the exact amount of outliers is not strictly needed to obtain a meaningful MeRCI score. Indeed, there is no brutal change around the correct outlier ratio (red line). Therefore in later experiments, we use the 95th percentile, giving some latitude to handle up to $5\%$ of possible outliers.

Table 1: Quantitative evaluation on the toy dataset

| Method | higher is better | | | lower is better | |
|---|---|---|---|---|---|
| | $Q_s$ | $L_s$ | $S_s$ | CRPS | MeRCI |
| Oracle | - | - | - | - | 19 |
| Constant | - | - | - | - | 81 |
| Multi Inits | -0.65 | -6.0 | 0.07 | 19.0 | 64 |
| Bagging | -0.02 | -4.8 | 0.13 | 16.8 | 53 |
| Monte Carlo Dropout | -0.03 | -4.9 | 0.11 | 17.1 | 60 |
| Multi Epochs | -28 | -6.9 | 0.0 | 19.3 | 81 |

## 6.4 RESULTS

Finally, we discuss the scores obtained for each uncertainty estimation technique. We expect Bagging as the top technique because it benefits from external sources of data. We also expect Multi Epochs to be the worst one, since in this setting, the network is likely to converge to a local minimum. Therefore there won't be much informative variance across epochs.

We show the results in Table 1. We also highlight the relative ranking of the different uncertainty estimation techniques with respect to each metric by using colors: from yellow for the best technique to red for the worst. For visibility the worst score is in gray. According to this table, all metrics including MeRCI agree on the ranking. As expected, they all classify Multi Epochs as the worst technique. Interestingly, this technique is clearly marked off by MeRCI as uninformative (obtaining as bad a score as a constant uncertainty prediction). Finally, the top techniques are Bagging and Monte Carlo Dropout with close scores according to every metric.

In a simple setting like this toy example, we demonstrate that all the classical metrics are well behaved in their ability to correctly rank different uncertainty estimation techniques. We also show that our proposed metric properly follows these tracks, being therefore ready for a more realistic evaluation setup.

## 7 ANALYSIS: THE REAL-WORLD TASK OF MONOCULAR DEPTH ESTIMATION

We now scale to real-world data with the study of a regression-based computer vision task.

### 7.1 EXPERIMENTAL PROTOCOL

This section presents experiments on depth estimation from monocular images on the NYU-Depth v2 indoor dataset Nathan Silberman & Fergus (2012). We use the set of 16K images proposed by Moukari et al. (2018). Also, we use their hourglass-type architecture, with a dilated ResNet200 pre-trained on ImageNet for the encoder part and 4 up-projection modules followed by a final convolution for the decoder. We then add dropout with probability 0.2 after the first 3 up-projection modules, in order to perform Monte-Carlo sampling at test time for the corresponding experiment.

### 7.2 RESULTS: QUALITATIVE EVALUATION

We first show a qualitative evaluation obtained for the monocular depth estimation, in the case where the predictor is common for each method. In Figure 4, we show an example of a test image 4(a), the prediction of the common predictor 4(b) and finally the absolute error between the prediction and the ground truth 4(c). Note that, according to the absolute error map, a few demarcated regions should be marked as uncertain: namely the carpet, the bedside lamp, the pillow and the window.

Table 2 exhibits the relative ranking of the methods for this specific image according to each metric. We see from the table that all state-of-the-art metrics globally agree on the ranking and designate Multi Network as being the top method. On the other hand, MeRCI puts Multi Networks at position 4. According to MeRCI, the most correlated predictive uncertainty method with the true error is Multi Epochs, while it is ranked from position 2 to 4 according to the other metrics. Moreover,

Table 2: Methods scores and ranking for a specific NYU Depth v2 testset image according to each metric

| Method | $Q_s$ | $L_s$ | $S_s$ | CRPS | MeRCI |
|---|---|---|---|---|---|
| Multi Inits | -3.4 | -1.9 | 0.87 | 0.12 | 0.40 |
| Bagging | -2.7 | -1.6 | 0.90 | 0.12 | 0.55 |
| Monte Carlo Dropout | -7.5 | -3.3 | 0.61 | 0.13 | 0.37 |
| Multi Epochs | -2.3 | -1.6 | 0.87 | 0.12 | 0.34 |
| Multi Networks | 1.8 | 0.25 | 1.3 | 0.10 | 0.41 |
| Learned Error | -67 | -3.1 | 0.80 | 0.13 | 2.3 |

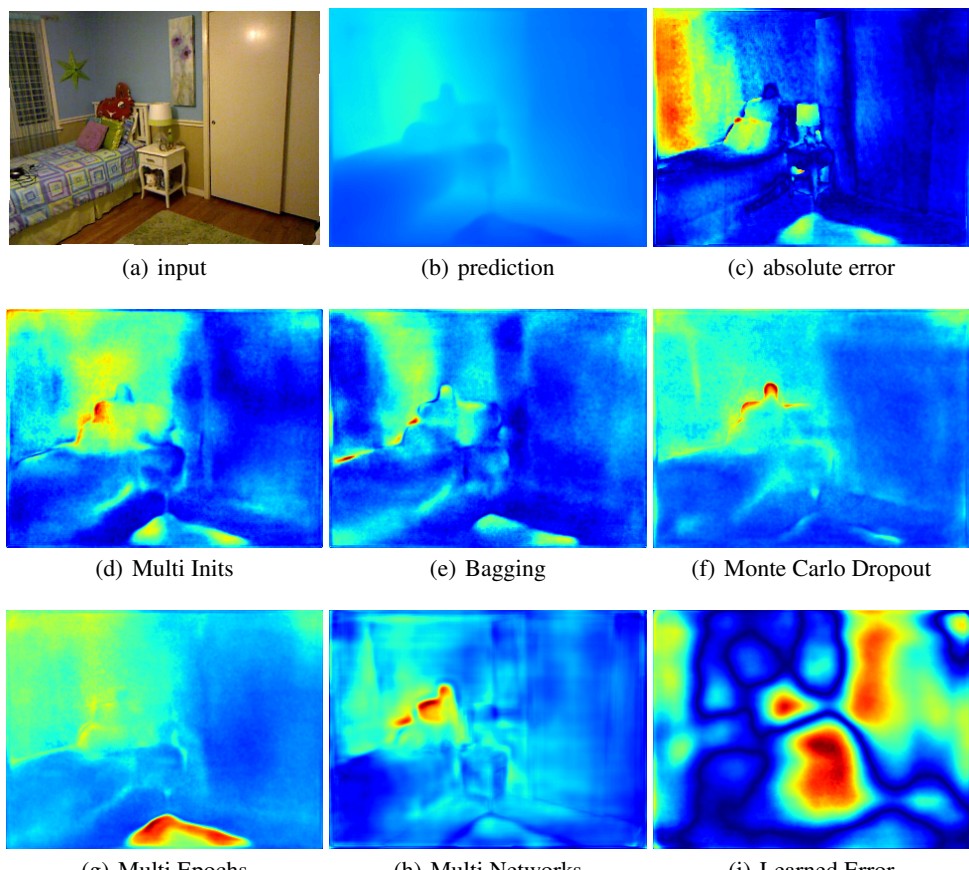

(a) input     (b) prediction     (c) absolute error

(d) Multi Inits     (e) Bagging     (f) Monte Carlo Dropout

(g) Multi Epochs     (h) Multi Networks     (i) Learned Error

Figure 4: A NYU Depth v2 test case. First row: the test image along with a predicted depth map and the corresponding absolute errors. Remaining rows: predictive uncertainties.

Monte Carlo Dropout has a bad ranking using the classical scoring rules (being number 5 or 6) while it is the second best method when using MeRCI. Unlike the toy example, we observe a clear disagreement between the various metrics. We look further to understand this discordance visually.

We display the 6 uncertainty estimates obtained with the different methods in Figure 4. Only two methods show high uncertainty in the region corresponding to the window : namely Multi Epochs and Monte Carlo Dropout. This region is clearly the most important as it is the the one with the highest errors according to the error map Figure 4(c). Furthermore, Multi Epochs also displays high uncertainty at the carpet location, while Monte Carlo Dropout misses this place. This is the reason why MeRCI ranked these methods as the two top ones, but favors Multi Epochs. Meanwhile, we see that Multi Networks, which is ranked first for all the common metrics, has yet poor correlation with the true error. Indeed, it shows high uncertainty around the pillow but misses all the other important

Table 3: Quantitative evaluation on the NYU Depth v2 test set. For comparison purpose, the depth prediction is the same for all the estimators.

| Method | higher is better | | | | | | | | | lower is better | | | MeRCI |
|---|---|---|---|---|---|---|---|---|---|---|---|---|---|
| | $Q_s$ | | | $L_s$ | | | $S_s$ | | | CRPS | | | |
| | U | G | L | U | G | L | U | G | L | U | G | L | |
| Oracle | - | - | - | - | - | - | - | - | - | - | - | - | 0.356 |
| Constant | - | - | - | - | - | - | - | - | - | - | - | - | 1.11 |
| MI | -3.1 | -3.0 | -4.3 | -4.7 | -3.4 | -2.8 | 0.56 | 0.56 | 0.50 | 0.32 | 0.32 | 2.5 | 1.22 |
| Bagging | -3.1 | -3.0 | -4.4 | -4.8 | -3.5 | -2.9 | 0.55 | 0.56 | 0.50 | 0.32 | 0.32 | 2.5 | 1.18 |
| MCD | -7.7 | -7.5 | -10 | -5.7 | -4.8 | -4.3 | 0.40 | 0.41 | 0.36 | 0.34 | 0.34 | 2.5 | 1.10 |
| ME | -2.9 | -2.7 | -4.1 | -4.9 | -3.7 | -3.0 | 0.54 | 0.54 | 0.48 | 0.32 | 0.32 | 2.5 | 0.91 |
| MN | -0.45 | -0.37 | -1.1 | -3.3 | -1.9 | -1.6 | 0.73 | 0.74 | 0.67 | 0.30 | 0.30 | 2.5 | 1.18 |
| LE | -137 | -134 | -169 | -5.2 | -4.3 | -3.8 | 0.46 | 0.46 | 0.41 | 0.33 | 0.33 | 2.5 | 4.31 |

spots. It also captures variance on the edges of the RGB image which is inconsistent to the true error. This is due to the fact that some of the different networks involved in the estimations produce sharper outputs but at the price of some artifacts transferred from the RGB images (Moukari et al., 2018). Eventually, although Learned Error completely fails to capture the relevant error information, some scoring rules do not rank it as the worst one while it is correctly classified by MeRCI. More visual results are presented in the appendix, including in particular images for which the depth ground truth are locally erroneous. This allows to analyze further how MeRCI handles these configurations.

### 7.3 Results: Quantitative evaluation for a common predictor

The quantitative results are presented in Table 3. As we want to see the impact of the different priors in the predicted distributions, we computed the analytical scoring rules and CRPS for three different probability density functions: a Uniform prior (**U**), a Gaussian prior (**G**) and a Laplace prior (**L**). We see that, despite the fact that the chosen priors have different decay behaviors, the choice of the probability density function does not influence a lot the relative ranking of the different methods. All these scores are based on predictive distributions, yet the form of the distribution does not have a major impact on the assessed quality of a method. On the other hand, MeRCI does not rely on any prior and is much more straightforward for judging different techniques.

Similarly to the single-image qualitative evaluation, Multi Networks is by far the best method with respect to all the classical state-of-the-art metrics. According to (Gneiting et al., 2007; Gneiting & Raftery, 2007), it means it has the best sharpness subject to calibration. However, MeRCI shows that this uncertainty estimation technique is less correlated with the true error than a constant (hence uninformative) uncertainty prediction.

Here again, MeRCI underlines Multi Epochs as the best forecast (in fact the only one to outperform a constant uncertainty prediction). This favored rank is supported by the fact that training a large network on a complex task and with a stochastic minimization scheme naturally yields oscillations around a minimum. What is more, along the last epochs, the network will focus on learning what is still improvable: i.e. regions with high errors. Thus, using the standard deviation of estimations obtained across epochs allows to capture these variations and highlights error-prone regions.

## 8 Conclusions

This paper has drawn a comprehensive review of the existing metrics used for evaluating predictive uncertainty and have analyzed the behavior of these metrics in regression tasks, firstly on a toy dataset and then on the real-world case of monocular depth estimation. We then proposed a novel metric, the Mean Rescaled Confidence Interval (MeRCI), also dedicated to the assessment of predictive uncertainty. We first showed that the proposed metric behaves healthily on the toy dataset, while giving latitude to be robust to outliers if needed (thanks to its percentile parameter). Then, we demonstrate that the proposed metric is closely related to the true error of the prediction and therefore more reliable than the previous ones, within the particular case of monocular depth estimation. Moreover, we saw that compared to previous methods, MeRCI has the advantage of being directly applicable to regression tasks with deep learning, without needing any prior on the distribution.

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

## Supplementary Material

### A    Results: Quantitative evaluation for proper predictors

Table 4: Quantitative evaluation on the NYU Depth v2 testset. For each method, we perform its own prediction and uncertainty.

| Method | higher is better | | | lower is better | |
| --- | --- | --- | --- | --- | --- |
| | $Q_s$ | $L_s$ | $S_s$ | CRPS | MeRCI |
| MI | -3.0 | -3.4 | 0.57 | 0.31 | 1.20 |
| Bagging | -3.0 | -3.5 | 0.55 | 0.32 | 1.19 |
| MCD | -7.7 | -5.0 | 0.37 | 0.37 | 1.20 |
| ME | -2.8 | -3.7 | 0.53 | 0.32 | 0.91 |
| MN | -0.63 | -2.3 | 0.66 | 0.32 | 1.20 |
| LE | -81 | -4.6 | 0.43 | 0.33 | 4.70 |

In Table 4 we show the quantitative evaluation on the NYU Depth v2 testset, where each method is used with its own couple prediction/uncertainty to compute the scores. Using proper predictors for each method allow us to compare the true scores of each technique, depending both on their calibration and sharpness. Indeed, fixing the prediction to be identical for each method bias the analysis of sharpness under calibration constraint: methods based on a mean of their estimations present different calibrations.

Because we show in Section 7.3 that the prior on the predictive distribution does not have a major impact on the ranking, we choose to only present results under a Gaussian prior. Results are very similar to the one presented in Table 3 with a common predictor. This means that, for this study, the proper calibrations computed for each method are close to each others. Hence, conclusions drawn in Section 7.3 remain the same.

# B  QUALITATIVE ANALYSIS

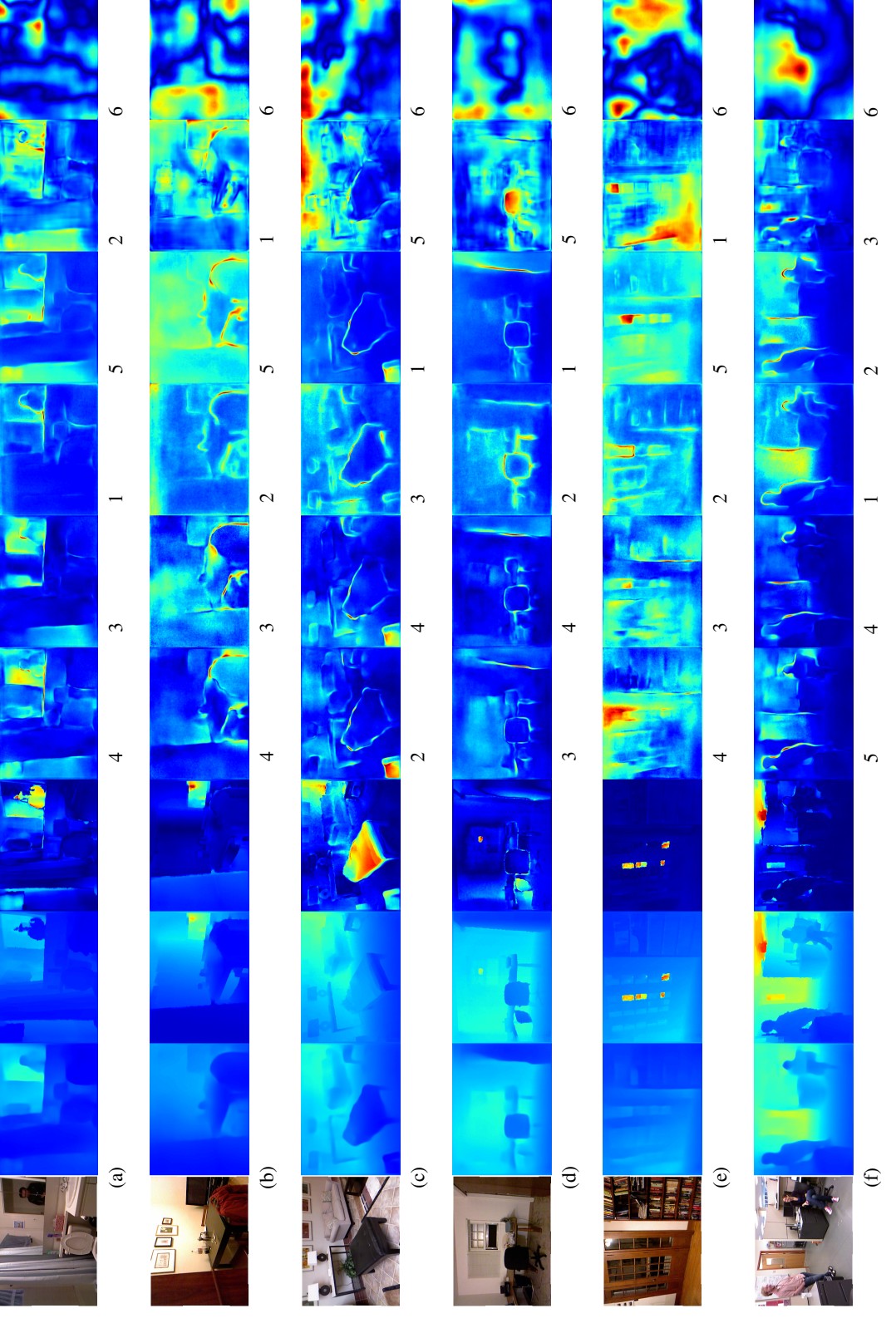

Figure 5: Qualitative analysis of several NYU Depth *v2* testset images (rows). Columns from left to right represent: input image, prediction, ground-truth, absolute error and MI, Bagging, MCD, ME, MN, LE uncertainty maps. For each example, the MeRCI scores are computed and the methods ranking are displayed under the uncertainty estimations.

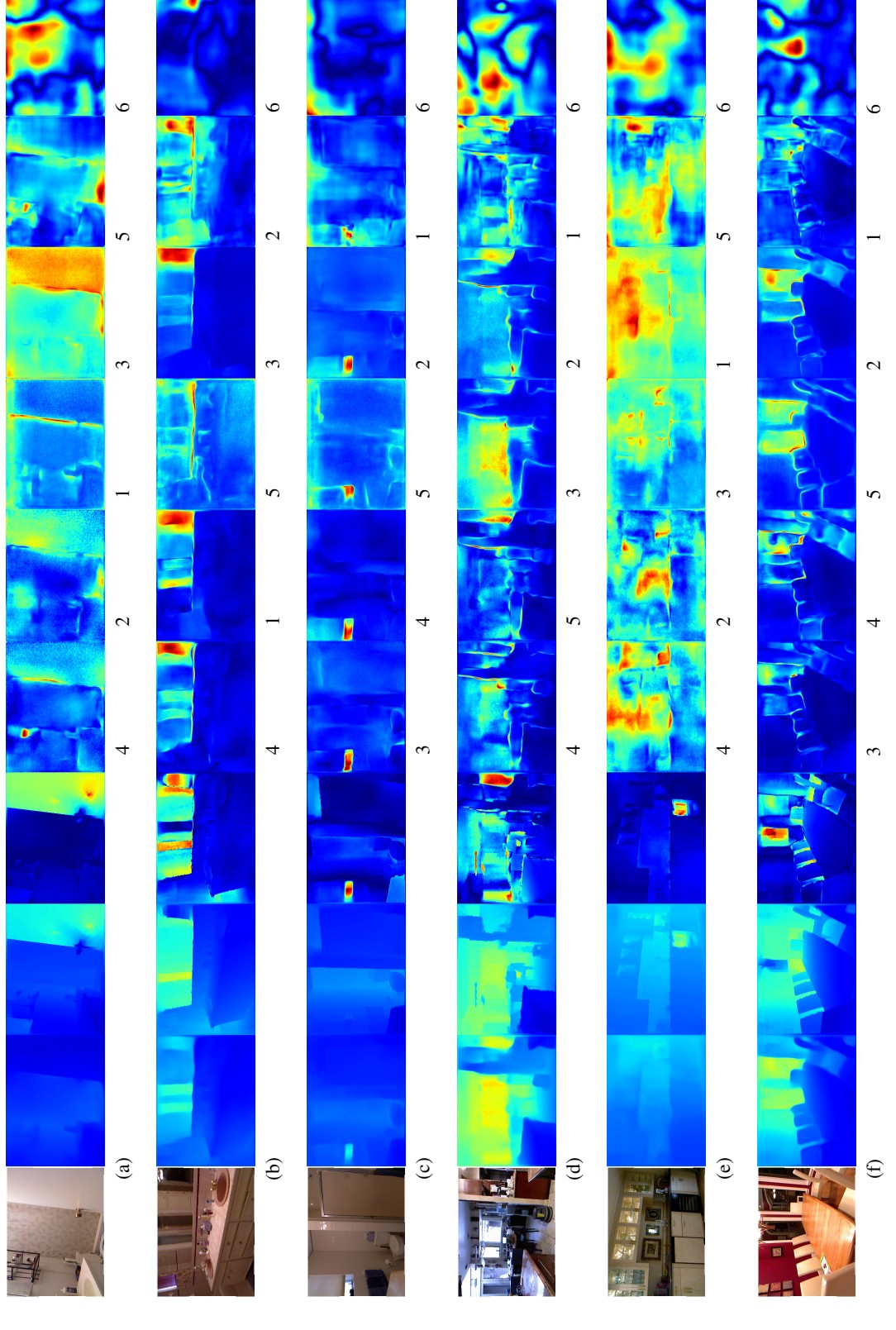

Figure 6: Qualitative analysis of several NYU Depth v2 testset images (rows). Columns from left to right represent: input image, prediction, ground-truth, absolute error and MI, Bagging, MCD, ME, MN, LE uncertainty maps. For each example, the MeRCI scores are computed and the methods ranking are displayed under the uncertainty estimations.

