# OpenReview forum: "MERCI: A NEW METRIC TO EVALUATE THE CORRELATION BETWEEN PREDICTIVE UNCERTAINTY AND TRUE ERROR"
_ICLR.cc/2019/Conference_

### Official Review · AnonReviewer2 · 2018-11-02
**Lacking novelty and rigor.**

**Rating:** 3
**Confidence:** 4

**Review:**

The main contribution of this paper is a new proposed score to evaluate models that yield uncertainty values for regression.

As constituted, the paper can not be published into one of the better ML conferences. The novelty here is very limited. Furthermore there are other weaknesses to the study.

First, the stated goal of the "metric" is that "reflects the correlation between the true error and the estimated uncertainty ... (and is) scale independent and robust to outliers." Given this goal (and the name of the paper) it is perplexing why the correlation (https://en.wikipedia.org/wiki/Pearson_correlation_coefficient) of true error and predicted error (\sigma_i) was not tried as baseline score. The correlation would have some "scale independence" and I'm sure that there are robust estimates (a simple thing would be to consider the median instead of mean, but there are probably more sophisticated approaches). This just feels like an obvious omission. If one wants to mix both predictive quality and correctness of uncertainty assessments then one could just scale the mean absolute error by the correlation: MAE/Corr, which would lead to a direct comparison to the  proposed MeRCI.

Second, the paper does a poor job of justifying MeRCI. On toy data MeRCI is justified by a confluence with existing scores. Then on the depth prediction task, where there are discrepancies among the scores, MeRCI is largely justified qualitatively  on a single image (Figure 4). A qualitative argument on a single instance in a single task can not cut it. The paper must put forth some systematic and more comprehensive comparison of scores.

Even with the above issues resolved, the paper would have to do more for publication. I would want to see either some proof of a property of the proposed score(s), or to use the proposed score to inform training, etc.

---

> ### Author Response · Authors · 2018-11-19
> **More analysis and experiments**
>
> Thank you for your time and critical feedback.
> The score you propose is indeed interesting and we would need to look further into it. However, it seems that its main disadvantage compared to our proposed metric is the lack of robustness to outliers. Indeed, there is no direct way to handle a known amount of outliers and the correlation itself is error-prone and will be sensitive to erroneous data.
>
> We believe that robustness to outliers is of major importance, especially for problems where the learning is based on potentially erroneous ground-truth. Depth estimation is one such typical case since it involves sensor measurement uncertainty. To highlight this issue and also to address your concern about our qualitative evaluation made on a single image, we plan to add several images in the appendix with and without ground truth problems.
>
> We finally would like to respectfully disagree concerning the novelty of the work since there is no reliable metric used in the deep learning community to assess the quality of a predictive uncertainty with respect to the true error, while the need for it is growing.

---

### Official Review · AnonReviewer3 · 2018-11-02
**Interesting paper for the deep learning community, but the experimental section is not convincing enough**

**Rating:** 5
**Confidence:** 3

**Review:**

This works presents an overview of different techniques to obtain uncertainty estimates for regression algorithms, as well as metrics to assess the quality of these uncertainty estimates.
It then introduces MeRCI, a novel metric that is more suitable for deep learning applications.

Being able to build algorithms that are good not only at making predictions, but also at reliably assessing the confidence of these predictions is fundamental in any application. While this is often a focus in many communities, in the deep learning community however this is not the case, so I really like that the authors of this paper want to raise awareness on these techniques. The paper is well written and I enjoyed reading it.
I feel that to be more readable for a broader audience it would be relevant to introduce more in depth key concepts such as sharpness and calibration, an not just in a few lines as done in the end of page 2.

While I found the theoretical explanation interesting, I feel that the experimental part does not support strongly enough the claims made in the paper. First of all, for this type of paper I would have expected more real-life experiments, and not just the monocular depth estimation one. This is in fact the only way to assess if the findings of the paper generalize.
Then, I am not convinced that keeping the networks predictions fixed in all experiments is correct. The different predictive uncertainty methods return both a mean and a variance of the prediction, but it seems that you disregard the information on the mean in you tests. If I understood correctly, I would expect the absolute errors to change for each of the methods, so the comparisons in Figure 4 can be very misleading.
With which method did you obtain the predictions in Figure 4.c?

Typos:
- "implies" -> "imply" in first line of page 3
- "0. 2" -> "0.2" in pag 6, also you should clarify if 0.2 refers to the fraction of units that are dropped or that are kept

---

> ### Author Response · Authors · 2018-11-19
> **More experiments**
>
> Thank you for your considerate review and insightful comments.
> About the concepts of calibration and sharpness, we will follow your advice and introduce them earlier in the paper to be more easily understandable for a broader audience.
>
> We also understand your concerns that the experiments do not support strongly enough our theoretical claims. As proposed by the reviewer 1, we plan to conduct similar experiments on several other datasets from the UCI benchmark. In this way, we should be able to demonstrate that our findings generalize well to other real-life setups.
>
> Finally, your last point is also very relevant. We indeed disregarded the means of the predictive uncertainty methods in our tests for comparison purposes. Instead, we use a unique prediction for all methods, represented in Figure 4. and obtained with the standard network described in section 7.1. However, we also computed all the results using both the mean and variance for each method to unbias the comparisons. Because the results were not mixed up, we originally decided to omit this part for simplicity and clarity. We realize that we should have left it and we will reintroduce the corresponding section in our next update.

---

> > ### Comment · AnonReviewer3 · 2018-11-23
> > **More experiments**
> >
> > Thanks for your reply. Without knowing the performance on the new experiments my opinion of the paper is however unchanged, so I will maintain the score as is.

---

### Official Review · AnonReviewer1 · 2018-11-03
**Ok, but not good enough**

**Rating:** 4
**Confidence:** 4

**Review:**

The authors propose mean rescaled confidence interval (MERCI) as a way to measure the quality of predictive uncertainty for regression problems. The main idea is to rescale confidence intervals, and use average width of the confidence intervals as a measure for calibration. Due to the rescaling, the MERCI score is insensitive to the absolute scale; while this could be a feature in some cases, it can also be problematic in applications where the absolute scale of uncertainty matters.

Overall, the current draft feels a bit preliminary. The current draft misses discussion of other relevant papers, makes some incorrect claims, and the experiments are a bit limited. I encourage the authors to revise and submit to a different venue.

There’s a very relevant ICML 2018 paper on calibrating regression using similar idea:
Accurate Uncertainties for Deep Learning Using Calibrated Regression
https://arxiv.org/pdf/1807.00263.pdf
Can you clarify if/how the proposed work differs from this? I’d also like to see a discussion of calibration post-processing methods such as Platt scaling and isotonic regression.

The paper unfairly dismisses prior work by making factually incorrect claims, e.g. Section 2 claims
“Indeed, papers like (Hernandez-Lobato & Adams, 2015; Gal & Ghahramani, 2016; Lakshminarayanan et al., 2017; Kendall & Gal, 2017) propose quantitative evaluations on several datasets, those classically used for evaluating the task, but only compare their average test performances in terms of RMSE. It is the quality of the prediction which is measured, and not the quality of the estimated uncertainty. They also show some qualitative results, where maps of the estimated uncertainty are displayed as images and visually evaluated. Yet, to the best of our knowledge, the literature on deep neural networks does not propose any method for the quantitative evaluation of the uncertainty estimates.”
This is incorrect.  To just name a few examples of prior work quantitatively evaluating the quality of uncertainty: (Hernandez-Lobato & Adams, 2015) and (Gal & Ghahramani, 2016) report log-likelihoods on regression tasks, (Lakshminarayanan et al. 2017) report log-likelihoods and Brier score on classification and regression tasks. There are many more examples.

The experiments are a bit limited. Figure 1 is a toy dataset and Table 2 / Figure 4 focus on a single test case which does not seem like a fair comparison of the different methods. The authors should at least compare their method to other work on the UCI regression benchmarks used by (Hernandez-Lobato & Adams, 2015).

---

> ### Author Response · Authors · 2018-11-19
> **More references and experiments**
>
> Thank you for your helpful review and constructive comments.
> Because we proposed a new evaluation metric, we were concerned by the question of how to compare very different predictive uncertainty. For example, the use of Bayesian deep learning techniques leads to compute variations that don’t have meaning in terms of absolute error. That is the reason why we took sides for the insensitivity to scale.
>
> Thank you also for pointing out this very relevant ICML paper that we were not aware of. We still need to study it further but it seems that they take into account calibration and sharpness separately, while we were driven by the will of assessing both these properties jointly. For example, the sharpness score they propose in section 3.5 seem very close to MeRCI as it corresponds to the mean variance of the evaluated method. However this only assesses the sharpness of the predictive uncertainty and do not allow to compare techniques with different ranges of uncertainties.
>
> We indeed made incorrect claims about some prior work and apologize for this mistake. We rather wanted to point out that systematic assessment of the uncertainty estimate itself was largely underestimated in the deep learning community. We will rephrase this properly in the next update.
>
> Finally we plan to use several other datasets from the UCI regression benchmark, as you advised. We believe that proving the generalization of the metrics behavior will support our claims more strongly. We will also add several other qualitative analysis in the appendix for a fairer comparison.

---

### Author Response · Authors · 2018-11-19
**Preface**

Foremost, we would like to thank the reviewers for their time and thorough reviews. Globally, all reviewers unveiled similar issues to the submitted work. We will address each specific point in detailed responses below. Just as a recall and to introduce our discussion, we would like to insist on the properties we wanted our metric to have:

       •	Insensitivity to scale, so that we can compare predictive uncertainty techniques with various ranges that are often not in relation with the absolute error;
       •	Robustness to outliers, so that we can fairly evaluate tasks for which data are known to be noisy;
       •	Easy applicability to deep learning, where we usually get two scalars to define the couple prediction/uncertainty instead of a whole distribution;
       •	Representativity of the true error, in the sense that we want it to evaluate jointly calibration and sharpness, ensuring that the estimated uncertainty varies with the true error.

We will also notify each submission updates we propose below.

---

### Meta-Review · Area_Chair1 · 2018-12-14

**Confidence:** 4
**Recommendation:** Reject

**Metareview:**

Reviewers are in a consensus and recommended to reject after engaging with the authors. Please take reviewers' comments into consideration to improve your submission should you decide to resubmit.